# A benchmark study of simulation methods for single-cell RNA sequencing data

Yue Cao[1,2], Pengyi Yang [ID] [1,2,3,4✉] & Jean Yee Hwa Yang [ID] [1,2,4✉]

Single-cell RNA-seq (scRNA-seq) data simulation is critical for evaluating computational methods for analysing scRNA-seq data especially when ground truth is experimentally unattainable. The reliability of evaluation depends on the ability of simulation methods to capture properties of experimental data. However, while many scRNA-seq data simulation methods have been proposed, a systematic evaluation of these methods is lacking. We develop a comprehensive evaluation framework, SimBench, including a kernel density estimation measure to benchmark 12 simulation methods through 35 scRNA-seq experimental datasets. We evaluate the simulation methods on a panel of data properties, ability to maintain biological signals, scalability and applicability. Our benchmark uncovers performance differences among the methods and highlights the varying difficulties in simulating data characteristics. Furthermore, we identify several limitations including maintaining heterogeneity of distribution. These results, together with the framework and datasets made publicly available as R packages, will guide simulation methods selection and their future development.

[1] Charles Perkins Centre, The University of Sydney, Sydney, Australia. [2] School of Mathematics and Statistics, The University of Sydney, Sydney, Australia. [3] Computational Systems Biology Group, Children's Medical Research Institute, Westmead, NSW, Australia. [4]These authors contributed equally: Pengyi Yang, Jean Yee Hwa Yang. ✉email: pengyi.yang@sydney.edu.au; jean.yang@sydney.edu.au

Single-cell RNA-sequencing (scRNA-seq) is a powerful technique for profiling the transcriptomes at the single-cell resolution and has gained considerable popularity since its emergence in the last decade[1]. To effectively utilise scRNA-seq data to address biological questions[2], the development of computational tools for analysing such data is critical and has grown exponentially with the increasing availability of scRNA-seq datasets. Evaluation of their performance with credible ground truth has thus become a key task for assessing the quality and robustness of the growing array of computational resources. While there exist certain control strategies such as spike-ins with known sequence and quantity, data that offer ground truth while reflecting the complex structures of a variety of experimental designs are either difficult or impossible to generate. Thus, in silico simulation methods for creating scRNA-seq datasets with desired structure and ground truth (e.g. number of cell groups) are an effective and practical strategy for evaluating computational tools designed for scRNA-seq data analysis.

To date, numerous scRNA-seq data simulation methods have been developed. The majority of these methods employ a two-step process of using statistical models to estimate the characteristics of real experimental single-cell data and using the learnt information as a template to generate simulation data. The distinctive difference between them is the choice of underlying statistical framework. Early methods often employ negative binomial (NB)[3–5] as it has been the typical choice for modelling gene expression count of RNA-seq[6]. Its variant, zero-inflated NB (ZINB) model takes account of excessive zeros in the count data and is chosen by other studies to better model the sparsity in single-cell data[7,8]. In more recent years, alternative models have been proposed with the aim to increase modelling flexibility including gamma-normal mixture model[9], beta-Poisson[10], gamma-multivariate hypergeometric[11] and the mixture of zero-inflated Poisson and log-normal Poisson distributions[12]. Other studies argued that parametric models with strong distributional assumption are often not appropriate to scRNA-seq data given its variability and proposed the use of a semi-parametric approach as the simulation framework[13]. Similarly, a recent deep learning-based approach[14] leverages the power of neural networks to infer underlying data distribution and avoid prior assumptions.

A common challenge of simulation methods is the ability to generate data that faithfully reflect experimental data[15]. Given that simulation datasets are widely used for the evaluation and comparison of computational methods[16], deviations of simulated data from properties of experimental data can greatly affect the validity and generalisability of the evaluation results. With the increasing number of scRNA-seq data simulation tools and the reliance on them to guide other method development as well as choosing the most appropriate data analytics strategy, a thorough assessment of all currently available scRNA-seq simulation methods is crucial and timely, especially when such an evaluation study is still lacking in the literature.

Here, we present a comprehensive evaluation framework, SimBench, for single-cell simulation benchmarking. Considering that realistic simulation datasets are intended to reflect experimental datasets in all data moments including both cell-wise and gene-wise properties, as well as their higher-order interactions, it is important to determine how well simulation methods represent all these values. To this end, we systematically compare the performance of 12 simulation methods across multiple sets of criteria, including accuracy of estimates for 13 data properties, the ability to retain biological signals and to achieve computation scalability, as well as their applicability. To ensure robustness of the results, we collect 35 datasets across a range of sequencing protocols and cell types. Moreover, we implement measure based on kernel density estimation[17] in the evaluation framework to enable the large-scale quantification and comparison of similarities between simulated and experimental data across univariate and multivariate distributions, and thus, avoid visual-based criteria which are often used in other studies. To assist development of new methods, we study potential factors affecting the simulation results and identify common strength and weakness of current simulation methods. Finally, we summarise the result into recommendation to the users, and highlight potential areas requiring future research.

## Results

**A comprehensive benchmark of scRNA-seq simulation methods on four key sets of evaluation criteria using diverse datasets and comparison measure.** Our SimBench framework evaluates 12 recently published simulation methods specifically designed for single-cell data (Fig. 1a, Table 1 and Supplementary Table 1). To ensure robustness and generalisability of the study results and account for variability across datasets (Supplementary Fig. 1), we curated 35 public scRNA-seq datasets (Fig. 1b and Supplementary Data 1) that include major experimental protocols, tissue types, and organisms. To assess a simulation method's performance on a given dataset, SimBench splits the data into input data and test data (referred to as the real data). Simulation data is generated based on the data properties estimated from the input data and compared with the real data in the evaluation process (Fig. 1c). Using four key sets of evaluation criteria (Fig. 1c, d), we systematically compare the single-cell simulation methods' performance for 432 simulation data representing 12 simulation methods applied to 35 scRNA-seq datasets.

The first set of evaluation criteria, termed data property estimation, aims to assess how realistic is a given simulated data. To address this, we first defined the properties for a given dataset with 13 distinct criteria and then developed a comparison process to quantify the similarity between the simulated and real data (Supplementary Fig. 2). The 13 criteria capture both the distributions of genes and cells as well as higher-order interactions, such as mean–variance relationship of genes. We anticipated that not all simulation methods will place emphasis on the same set of data properties and it is thus important to incorporate a wide range of criteria. We then examined a number of statistics for measuring distributional similarity[18]. Supplementary Fig. 3 shows that all statistics show similar performance with mean correlation of 0.7 and we have chosen to use the kernel density based global two-sample comparison test statistic[17] (KDE statistic), in our current study as it is applicable to both univariate and multivariate distributions.

The other three sets of evaluation criteria seek to assess each simulation method's ability to maintain biological signals and computational scalability and its applicability. For biological signals, we measured the proportion of differentially expressed (DE) genes obtained in the simulated data using DE detection methods designed for bulk and single-cell RNA-seq data, as well as four other types of gene signals of differentially variable (DV), differentially distributed (DD), differential proportion (DP) and bimodally distributed (BD) genes (see 'Methods'). A similar proportion to the real data would indicate an accurate estimation of biological signals present in the data. Scalability reflects the ability of simulation methods to efficiently generate large-scale datasets. This is measured through computational run time and memory usage with respect to the number of cells. Applicability examines the practical application of each method in terms of whether it can estimate and simulate multiple cell groups and allow simulation of differential expression patterns. Overall, our framework provides recommendations by taking into account all aspects of evaluation (Fig. 1e).

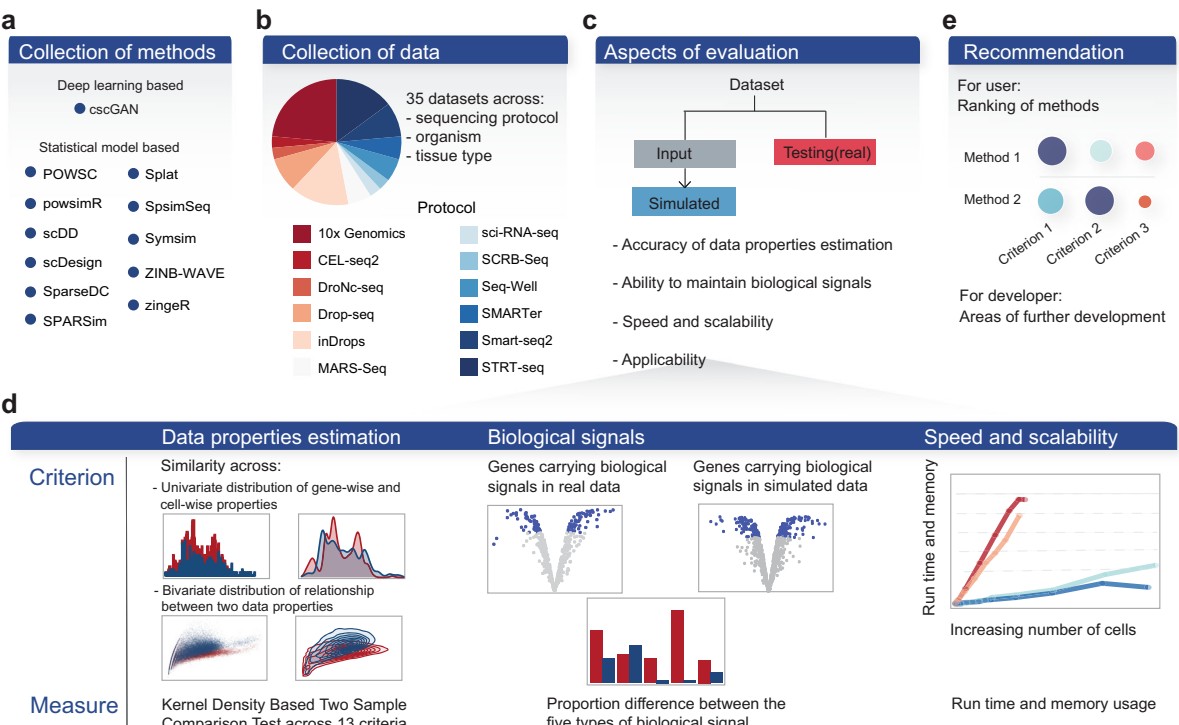

**Fig. 1 Schematic of the benchmarking workflow. a** A total number of 35 datasets, covering a range of protocols, tissue types, organisms and sample size was used in this benchmark study. **b** We evaluated 12 simulation methods available in the literature to date. **c** Multiple aspects of evaluation were examined in this study, with the primary focuses illustrated in detail in panel **d**. **e** Finally, we summarised the result into a set of recommendations for users and identified potential areas of improvement for developers.

**Comparison of simulation methods revealed their relative performance on different evaluation criteria.** Through ranking the 12 methods on the above four sets of evaluation criteria, we found that no method clearly outperformed other methods across all criteria (Fig. 2). We therefore examined each set of criteria individually in detail below and the variability in methods' performance within and across the four sets of evaluation criteria.

For data property estimation, we observed variability in methods' performance across the 13 criteria. ZINB-WaVE, SPARSim and SymSim are the three methods that performed better than the others across almost all 13 data properties (Fig. 2a). For the remaining methods, a greater discrepancy was observed between the 13 criteria, in which the rankings of methods based on each criterion do not show any particular relationship or correlation structure. Overall, our results highlight the relative strengths and weaknesses of each simulation method on capturing the data properties.

We observed that some methods (e.g. zingeR and scDesign) that were not ranked the highest in data properties estimation performed well in retaining biological signals (Fig. 2b). scDesign is designed for the purpose of power calculation and sample size estimation, while zingeR is designed to evaluate the DE detection approach in its publication and thus both methods require an accurate simulation and estimation of biological signals, particularly differential expression. It is not unexpected that they ranked highly in this aspect despite not being the most accurate in estimating other data properties.

For computational scalability, the majority of methods showed good performance with runtime of under 2 h and memory consumption of under eight gigabytes (GB) (Supplementary Fig. 4) when tested on the downsampled Tabula Muris dataset[19] with 50–8000 cells (see 'Methods'). However, some top performing methods, such as SPsimSeq and ZINB-WaVE revealed poor scalability (Fig. 2c). This highlights the potential trade-off

between computational efficiency and complexity of modelling framework. SPsimSeq, for example, involves the estimation of correlation structure using Gaussian-copulas model and scored well in maintaining gene- and cell-wise correlation. Its advantage came at the cost of poor scalability, taking nearly 6 h to simulate 5000 cells. Thus, despite the ability to generate realistic scRNA-seq data, the usefulness of such methods may be partially limited if a large-scale simulation dataset is required. In contrast, methods such as SPARSim, which was ranked second in parameter estimation as well as being one of top tier methods in scalability, may better suit needs if a large-scale simulation dataset is required by users.

Lastly, we found that different simulation methods satisfy different numbers of the applicability criteria (Fig. 2d). This is due, in part, to the fact that not all simulation methods are designed as general purpose simulation tools. For example, powsimR was originally designed as a power analysis tool for differential expression analysis but was included as a simulation tool by a number of simulation studies[9,10] in their performance comparison with other simulation methods. Being a power analysis tool, its primary usage is to simulate two cell groups from a homogenous cell population with a user-defined amount of differential expression. In contrast, a number of other methods such as SPARSim, SymSim and Splat that are originally intended as general purpose simulation tools are able to simulate multiple cell groups with user-defined differential expression patterns. We have outlined the primary purpose and the limitations of each method on this front in more detail in Table 1 to guide users in making informed decisions on methods that best suited to their needs.

**Impact of data- and experimental-specific characteristics on model estimation.** Aside from comparing the overall performance of methods to guide method selection, it is also necessary

**Table 1 scRNA-seq simulation methods evaluated in this study.**

| Methods | Year of publication | Approach | Estimate from multiple cell groups | Simulate multiple cell groups | Customise DE expression[a] | Assign gene name to generated data | Primary purpose as general simulation? |
|---|---|---|---|---|---|---|---|
| scDD[5] | 2016 | Dirichlet process mixture of normals | Restricted to two groups | Restricted to two groups | Yes | No | No, used for generating differentially distributed genes defined in the scDD study and evaluating the scDD framework |
| Splat[4] | 2017 | Gamma distribution for modelling mean expression; Poisson distribution for modelling count | No, requires a homogenous population (e.g. one cell type) | Yes, can simulate any number of groups | Yes | No | Yes |
| powsimR[3] | 2017 | Negative binomial or zero-inflated negative binomial model | No, requires a homogenous population (i.e. one cell type) | Restricted to two groups | Yes | Yes | No, power analysis tool for single-cell and bulk RNA-seq |
| SparseDC[29] | 2017 | Optimisation framework | Restricted to two conditions with multiple cell groups within each condition | Restricted to two conditions with multiple cell groups within each condition | Yes | No | No, used for generating the simulation data for assessing the performance of the SparseDC clustering method |
| zingeR[8] | 2018 | Negative binomial model with additive logistic regression to account for zeros | Yes, can estimate from any number of groups | Yes, can simulate any number of groups | Yes | No | No, used for generating simulation data for assessing the performance of the zingeR DE method |
| ZINB-WaVE[7] | 2018 | Zero-inflated negative binomial model | Yes, can estimate from any number of groups | Restricted to the groups in the input data | No | No | No, dimension reduction method for scRNA-seq |
| SymSim[10] | 2019 | Kinetic model using Markov chain Monte Carlo | No, requires a homogenous population (i.e. one cell type) | Yes, can simulate any number of groups | Yes | No | Yes |
| scDesign[9] | 2019[b] | Gamma-normal mixture model | Restricted to one and two groups | Restricted to one and two groups | Yes | No | No, power analysis tool for scRNA-seq |
| SPARSim[11] | 2020 | Gamma distribution for modelling expression; multivariate hypergeometric distribution for modelling technical variability | Yes, can estimate from any number of groups | Yes, can simulate any number of groups | Yes | Yes | Yes |
| SPsimSeq[13] | 2020 | Estimation of probability distribution uses fast log-linear model-based density estimation method; Gaussian- | Yes, can estimate from any number of groups | Restricted to the groups in the input data | Yes | Yes | Yes |

**Table 1 (continued)**

| Methods | Year of publication | Approach | Estimate from multiple cell groups | Simulate multiple cell groups | Customise DE expression[a] | Assign gene name to generated data | Primary purpose as general simulation? |
|---|---|---|---|---|---|---|---|
| | | copulas for modelling gene-gene correlation | | | | | |
| POWSC[12] | 2020 | Mixture of zero-inflated Poisson for modelling inactive transcription; log-normal Poisson for modelling the active transcription | Yes, can estimate from any number of groups | Restricted to the groups in the input data | Yes | No | No, power analysis tool for scRNA-seq |
| cscGAN[14] | 2020 | Generative adversarial network with Wasserstein distance | Yes, can estimate from any number of groups | Restricted to the groups in the input data | No | Yes | Yes |

[a]Includes either proportion of differential expression or fold change.
[b]We benchmarked the version of scDesign published in 2019. We note that during the final preparation stage of our work, a newer version scDesign2 was published[35].

to identify specific factors influencing the outcome of simulation methods. Here, we examined the impact of data- and experimental-specific characteristics including cell numbers and sequencing protocols on simulation model estimation.

To explore the general relationship between cell number and accuracy of data property estimation across simulation methods, we evaluated each method on thirteen subsamples of Tabula Muris data with varying numbers of cells but fixed number of cell types (see 'Methods'). Through regression analysis, we found certain data properties such as mean–variance relationships were more accurately estimated under datasets with larger numbers of cells, as shown by the positive regression coefficients (Fig. 3a and Supplementary Fig. 5). Nevertheless, most other data properties in the simulated data were negatively correlated with the increasing number of cells (e.g. library size, gene correlation). These observations suggest that overall, the increasing cell number may be accompanied by the increasing complexity of data and thus maintaining data properties may become more challenging. Future method development should consider this factor as an aspect of evaluation when assessing model performance.

To examine the impact of sequencing protocols, we utilised datasets consisting of multiple protocols applied to the same human PBMC and mouse cortex samples from the same study[20]. Figure 3b and Supplementary Fig. 7 reveal no substantial impact was introduced by protocol difference on the overall simulation results, as indicated by the flatness of the line representing the accuracy of each data property across each protocol. Taken together, these results indicate that the choice of reference input being shallow sequencing or deep sequencing has no substantial impact on the overall simulation results. Given that SymSim and powsimR are the only two methods that require specification of input data as either deep or shallow protocols, these results suggest that a general simulation framework for the two major classes of protocols may be sufficient.

**Comparison across criteria revealed common areas of strength and weakness**. While the key focus of our benchmark framework is assessing methods' performance across multiple criteria, we can further use these results to identify criteria where most methods performed well or were lacking (Fig. 4a). Comparing across criteria, those that display a large difference between the simulated and real data for most methods are examples of common weakness. This ability to identify common weakness has implications for future method development as it highlights ongoing challenges of simulation methods.

First, we compared the accuracy of maintaining each data property, where a larger KDE score indicates greater similarity between simulated and real data. Figure 4b shows data properties relating to the higher-order interactions including mean–variance relationship of genes revealed larger differences between the simulated and real data. In comparison, a number of gene- and cell-wise properties such as fraction of zero per cell had relatively higher KDE scores, suggesting they were more accurately captured by almost all simulation methods. These observations thus highlight the difficulty in incorporating higher-order interactions by current simulation methods in general, and the potential area for method development.

The ability to recapture biological signals was quantified using the metric symmetric mean absolute percentage error (SMAPE), where a score closer to 1 indicates greater similarity between simulated and real data (see 'Methods'). We found differentially distributed (DD) and differential proportion (DP) genes exhibited a greater difference between simulated and real data (Fig. 4b). We also noted that four out of the 12 methods consistently had very

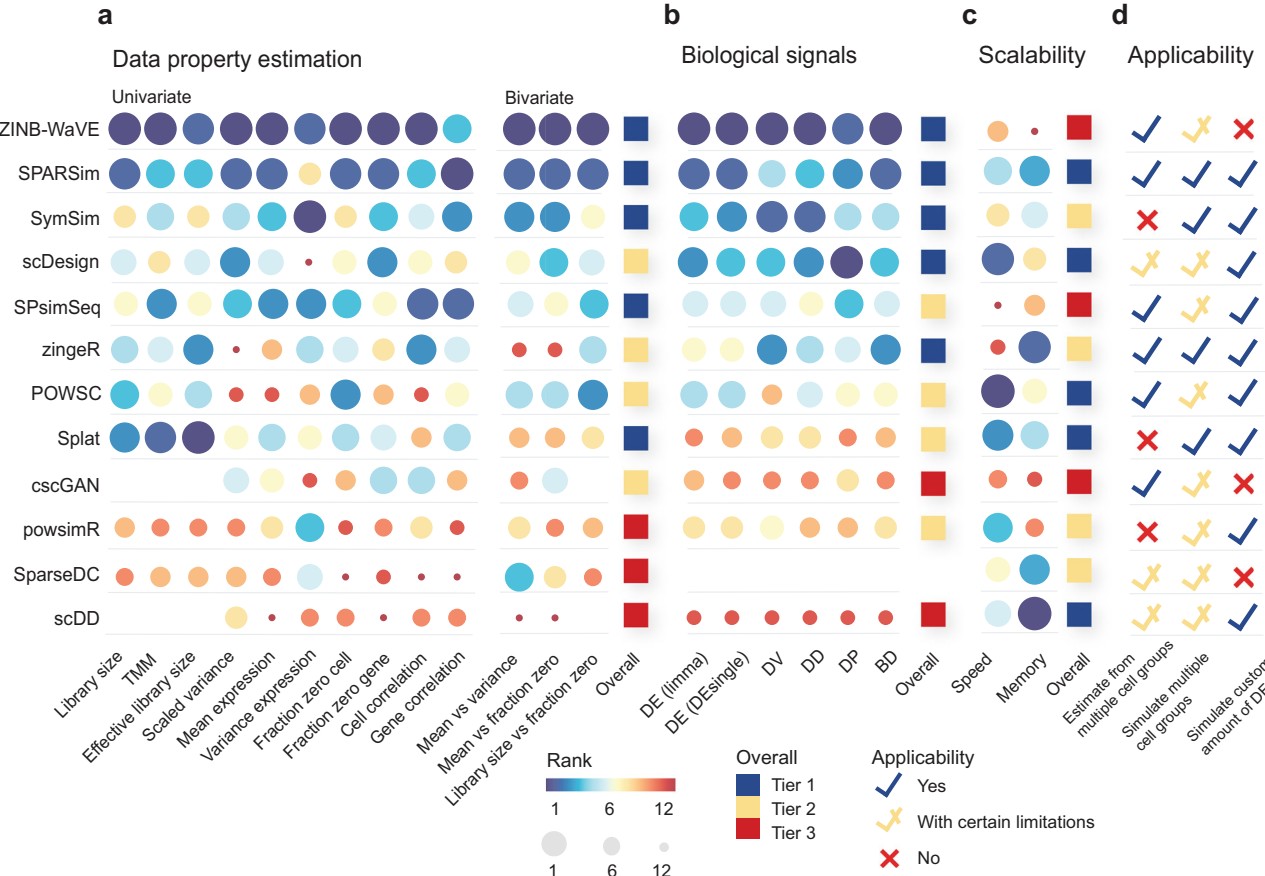

**Fig. 2 Ranking of methods across key aspects of evaluation criteria.** The colour and size of the circle denote ranking of methods, where a large blue circle represents the best possible rank of 1. Missing space indicates where a measurement was not able to be obtained, for example, due to the output format being normalised count instead of raw count (see 'Methods'). The ranks within each criterion were summarised into an overall tier rank, with tier 1 being the best tier. **a** Ranking of methods within data property estimation, ranked by median score across multiple datasets. **b** Ranking of methods within biological signals, ranked by median score across multiple datasets. **c** Scalability was ranked by the total computational speed and memory usage required for properties estimation and dataset generation across datasets. **d** Applicability was examined in terms of three criteria, which are explained in more detail in Table 1. The number of datasets used in the entire evaluation process and the success rate of each method on running the datasets is reported in Supplementary Fig. 4.

low SMAPE score of between 0 and 0.3, indicating the biological signals in the simulated data were at a very different proportion to that in real data. Upon closer examination, these methods simulated close to zero proportions of biological signals irrespective of the true proportion in the real data (Supplementary Fig. 6). Together, these observations point to the need for better strategies to simulate biological signals.

**Discussion**

We presented a comprehensive benchmark study assessing the performance of 12 single-cell simulation methods using 35 datasets and a total of 25 criteria across four aspects of interest. Our primary focus was on assessing accuracy of data property estimation and various factors affecting it, ability to maintain biological signals and computational scalability, as well as applicability. Additionally, using these results we also identified common areas of strength and weakness of current simulation tools. Altogether, we highlighted recommendations for method selection and identified areas of improvement for future method development.

We found that various underlying models were used for different simulation methods (Table 1). Each of the five top performing methods in category 1, for instance, uses a different underlying statistical approach (Table 1). As another example, the

three methods ZINB-WaVE, zingeR and powsimR differ substantially in detail despite the fact that they are all inspired by representing the observed counts using the NB family. Specifically, zingeR uses NB distribution to fit the mean and dispersion of the count data and model the excess zero using the interaction between gene expression and sequencing depth using additive logistic regression model. powsimR uses the standard ZINB distribution to fit the mean and dispersion of the count data, with the zero inflation modelled using binomial sampling. In ZINB-WaVE, the ZINB distribution is used to fit the mean and dispersion of the count data, as well as the probability that a zero is observed. Additionally, the estimation of mean and zero probability incorporates an additional parameter adapted from the RUV framework[21] to capture unknown cell-level covariates. Therefore, while both powsimR and ZINB-WaVE use ZINB distribution to fit the count data, the actual model differs. Interestingly, while deep learning methods have dominated various fields and applications, cscGAN, a deep learning-based model, for scRNA-seq data simulation only had moderate performance compared to the other models. This may be due to the large number of cells required for training the deep neural network in cscGAN as was demonstrated in their original study[14].

Based on the experiments conducted, we identified several areas of exploration for future researchers. Maintaining a

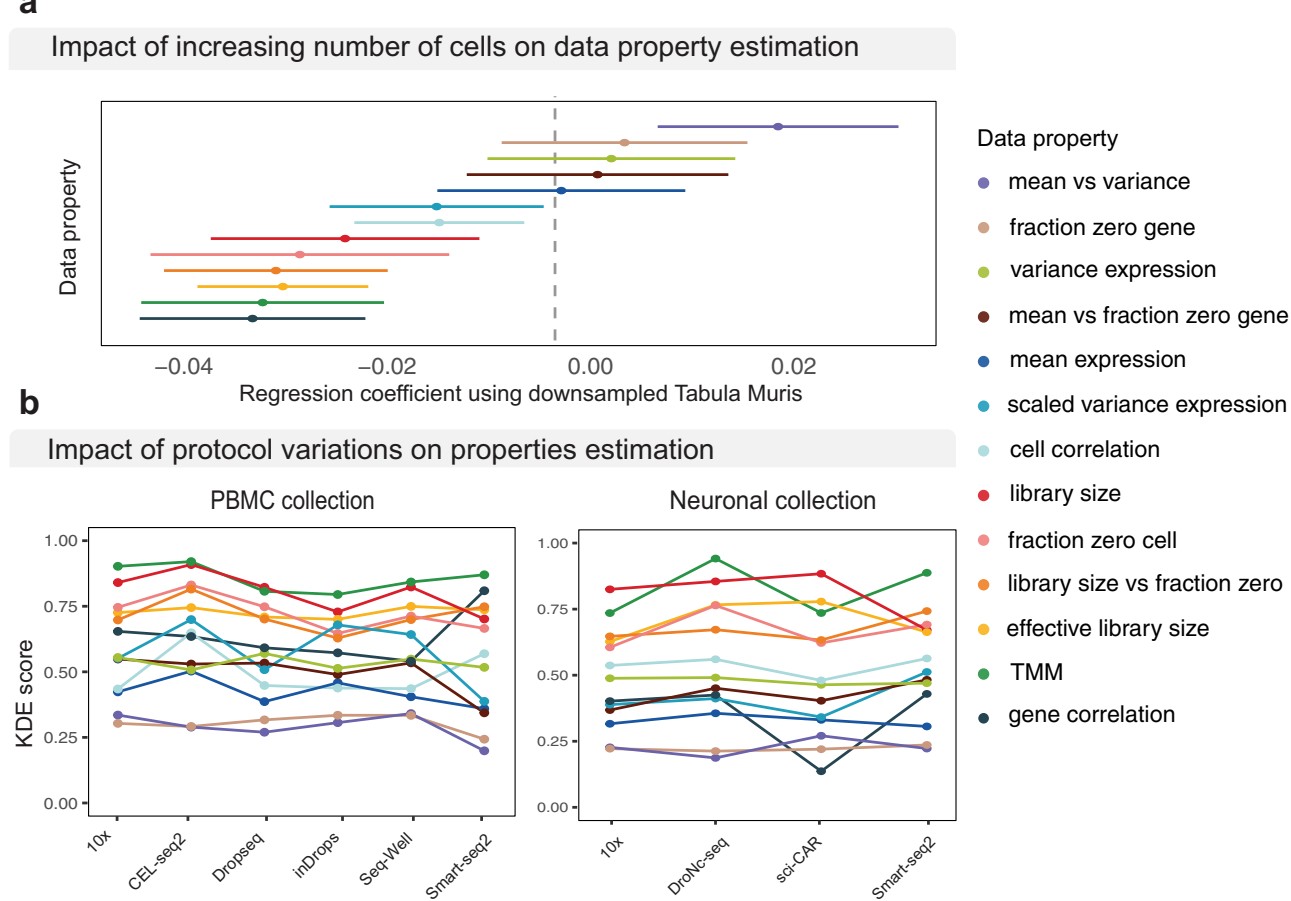

**Fig. 3 Impact of dataset characteristic on method performance. a** Impact of the number of cells on selected properties (see Supplementary Fig. 6 for all properties). Line shows the trends with increasing cell numbers. Dot indicates where a measurement is taken. **b** Impact of protocols was examined using two collections of datasets (see Supplementary Fig. 7 for individual methods). Boxplots show the individual score of each property for each method.

reasonable amount of biological signal is desirable and was not well captured by a number of methods. We also observed the genes generated by some methods (Table 1) were assigned uninformative names such as gene 1 and exhibit no relationship with genes from the real data. This limited us to assessing the proportion of biological signals in the simulated data, instead of assessing whether the same set of genes carrying biological signals (e.g. marker gene) are maintained in the simulated data. Incorporating the additional functionality of preserving biologically meaningful genes is likely to increase the usability of future simulation tools. Furthermore, we noted that several simulation studies only assessed their methods based on a number of gene- and cell-wise properties and did not examine higher-order interactions. Those studies are thus limited in the ability to uncover limitations in their methods. In comparison, our benchmark framework covered a comprehensive range of criteria and identified relative weakness of maintaining certain higher-order interactions compared to gene- and cell-wise properties.

As expected, we identified that none of the simulation methods assessed in this study could maintain the heterogeneity in cell population that was due to patient variability. This is potentially related to the paradigm used by current simulation techniques, as some methods implicitly require input to be a homogeneous population. For instance, some simulation studies inferred modelling parameters and performed simulation on each cell type separately when the reference input contains multiple cell types. However, experimental datasets with data from multiple samples, for example multiple patients, would be characterised by sample-to-sample variability within a cell type. This cellular heterogeneity is an important characteristic of single-cell data and has key applications such as identification of subpopulations. The loss of heterogeneity can thus be a limiting factor, as in some cases the simulation data could be an oversimplified representation of single-cell data. Future research such as phenotype-guided simulation[22] can help to extend the use of simulation methods.

Finally, we found there exists various trade-offs between the four aspects of criteria and having a well-rounded approach could be more important than a framework that performs best on one aspect but limiting in the other aspects. For example, as single-cell field advances and datasets with hundreds of thousands of cells become increasingly common, users may be interested in simulating large-scale datasets to test the scalability of their methods. As a result, methods that rank highly on scalability while also performing well on other aspects (e.g. SPARSim, scDesign and Splat) may be more favourable than other methods under these scenarios. We also note that due to the primary intended purpose of each method, not all methods allow users to customise the number of cell groups and the amount of differential expression between groups. Method that offers a well-rounded approach across multiple aspects of interests is therefore a direction of future research.

While we aim to provide a comprehensive assessment of available simulation methods, our study is not without limitations. For example, a few methods were excluded in this study due to their unique properties. SERGIO[23] is able to simulate regulation of genes by transcriptional factors, and therefore

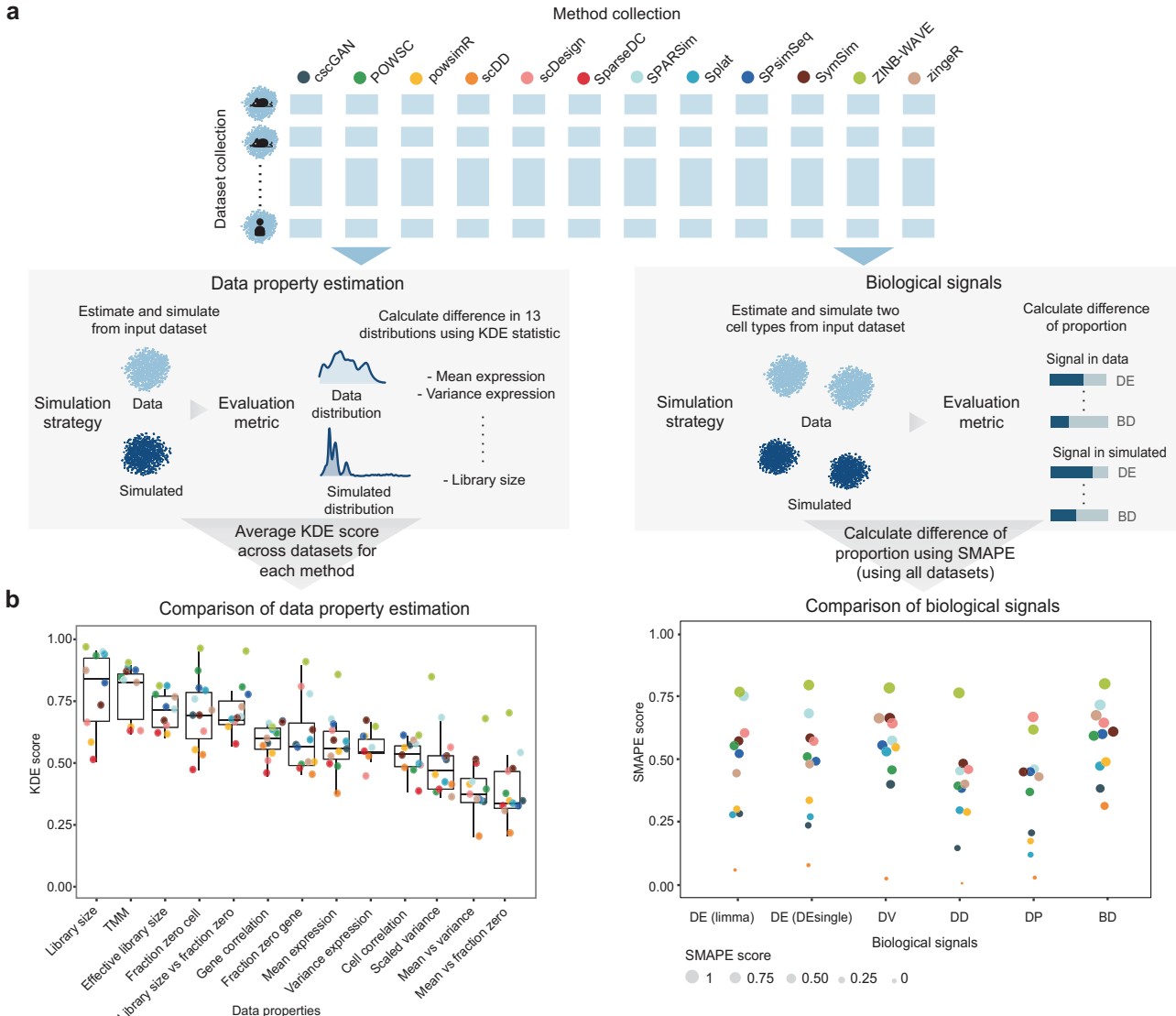

**Fig. 4 Comparison of criteria in data property estimation and in biological signals. a** Evaluation procedure for data property estimation and biological signals. **b** The evaluation results and the comparison of criteria within the two aspects of evaluation. For data property estimation, the KDE score measures the difference between the distribution of 13 data properties in simulated and in real data. A score close to 1 indicates a greater similarity. Each boxplot shows the distribution of the median KDE score attained by all simulation methods ($n = 12$), with the KDE score attained by each method shown in individual data point. The box represents quartiles, the line represents the median, the lower and upper whisker represents the bottom 25% and top 25% of the data. Outliers can be seen from the individual data points that are outside the whiskers. For biological signals, the SMAPE score measures the percentage difference between the proportion of biological signals detected in simulated and in real data. A score of 1 indicates no difference in the biological signals detected in real and simulated data and a score of 0 indicates maximal difference.

requires gene regulatory networks as one of the inputs. Both PROSSTT[24] and dyngen[25] are designed to simulate scRNA-seq data with trajectory information and require user-defined lineage trees. Lun's[26] was originally designed to tackle confounding plate effects in DE analysis and it requires plate information to be specified in the input. These simulation methods may need special considerations and evaluation criteria that could not be captured by the general framework in this study. Although the choice of DE detection methods could affect the evaluation of the simulation methods, our evaluation using both limma, a DE method originally designed for bulk RNA-seq data, and DEsingle, a DE method specifically designed for scRNA-seq data demonstrate a high agreement of the rankings of simulation methods based on the two DE methods (Fig. 2b).

In conclusion, we have illustrated the usefulness of our framework by summarising each method's performance across different

aspects to assist with method selection for users and identify areas of further improvement for method developers. We advise users to select the method that offers the functionality best suited to their purpose and developers to address the limitations of current methods. The evaluation framework has been made publicly available as the R package SimBench (https://github.com/SydneyBioX/SimBench). SimBench allows any new simulation methods to be readily assessed under our framework. It requires two inputs including the simulated data generated by any simulation method and the real data that was used as the reference input to generate the simulated data. SimBench then runs the evaluation procedure as performed in this study. We also provide all datasets used in this study as a Bioconductor data package SimBenchData (https://bioconductor.org/packages/devel/data/experiment/html/SimBenchData.html). Together these two packages enable future simulation methods to be assessed

and compared with the methods benchmarked in this study. Additionally, we provide a Shiny application for interactively exploring the results presented in this study hosted at http://shiny.maths.usyd.edu.au/. The application allows users to select datasets of their interest, such as within a certain range of cell numbers, and examine methods performance based on the specified datasets. Furthermore, we will provide updates to the website to include the benchmark results from new simulation methods when they become available so that our comparative study will stay up-to-date and will support future method development.

## Methods

**Dataset collection.** A total of 35 publicly available datasets was used for this benchmark study. For all datasets, the cell type labels are either publicly available or obtained from the authors upon request[27]. Details of each dataset including their accession code are included in the Supplementary Data 1. The datasets contain a range of sequencing protocols including both unique molecular identifiers (UMIs) and read-based protocols, multiple tissue types and conditions, and from human and mouse origin.

The raw (unnormalised) count matrix was obtained from each study and quality control was performed by removing potentially low-quality cells or empty droplets that expressed less than one percent of UMIs. For methods that require normalised count, we converted the raw count into log2 counts per million reads (CPM), with addition of pseudocount of 1 to avoid calculating log of zero.

Note the Tabula Muris dataset was only used to benchmark speed and scalability of methods. Properties estimation was evaluated on all other datasets. For evaluating biological signals, 25 datasets containing multiple cell types or conditions as specified by Supplementary Data 1 were used.

**Selection and implementation of simulation methods.** An extensive literature review was conducted and a total of 12 published single-cell simulation methods with implementation available in R and Python was found. The details of each method, including the version of the code used in this benchmark study and its publication are outlined in Table 1 and Supplementary Table 1. Supplementary Table 2 detailed the execution strategy of each method for data property estimation and biological signals and is dependent on the input requirement and the documentation of each method. Where possible, default setting or suggested setting from documentation is followed.

To ensure the simulated data is not simply a memorisation of the original data, we randomly split each dataset into 50% training and 50% testing (referred to as the real data in this study). The training data was used as input to estimate model parameters and generate simulated data. The real data was used as the reference to evaluate the quality of the simulated data, by comparing the similarity between the simulated data and the real data. The same training and testing subset was used for all methods to avoid the data splitting process being a confounding factor in evaluation.

All methods were executed using a research server with dual Intel(R) Xeon(R) Gold 6148 Processor (40 total cores, 768 GB total memory). For methods that support parallel computation, we used 8 cores and stopped the methods if the simulation was not completed within 3 h. For methods that run on a single core, we stopped the methods if not completed within 8 h.

### Evaluation of data property estimation

*Data properties measured in this study.* We adapted the implementation from countsimQC (v1.6.0)[18], which is an R package developed to evaluate the similarities between two RNA-seq datasets, either bulk or single-cell and evaluated a total of 13 data properties across univariate and bivariate distribution. They are described in detail below:

- Library size: total counts per cell.
- TMM: weighted trimmed mean of $M$-values normalisation factor[28].
- Effective library size: library size multiplied by TMM.
- Scaled variance: $z$-score standardisation of the variance of gene expression in terms of log2 CPM.
- Mean expression: mean of gene expression in terms of log2 CPM.
- Variance expression: variance of gene expression in terms of log2 CPM.
- Fraction zero cell: fraction of zeros per cell.
- Fraction zero gene: fraction of zeros per gene.
- Cell correlation: Spearman correlation between cells.
- Gene correlation: Spearman correlation between genes.
- Mean vs variance: the relationship between mean and variance of gene expression.
- Mean vs fraction zero: the relationship between mean expression and the proportion of zero per gene.
- Library size vs fraction zero: the relationship between library size and the proportion of zero per gene.

Note that properties relating to library size, including TMM and effective library size can only be calculated using unnormalised count matrix and could not

be obtained from methods that generate normalised count. As a result, these scores were shown as a blank space in all relevant figures.

*Evaluation measures.* In this study, we used a non-parametric measure termed kernel density based global two-sample comparison test[17] (KDE test) to compare the data properties between simulated and real data. The discrepancy between two distributions is calculated based on the difference between the probability density functions, either univariate or multivariate, which are estimated via kernel smoothing.

The null hypothesis of the KDE test is that the two kernel density estimates are the same. An integrated squared error (ISE) serves as the measure of discrepancy and is subsequently used to calculate the final test statistic under the null hypothesis. The ISE is calculated as:

$$T = \int [f_1(x) - f_2(x)]^2 \, dx \tag{1}$$

where $f_1$ $f_1$ and $f_2$ are the kernel density estimates of sample 1 and sample 2, respectively. The implementation from the R package *ks* (v1.10.7) was used for the KDE test performed in this study.

We used the test statistic from the KDE test as the measure to quantify the extent of similarity between simulated and real distributions. We applied a transformation rule by scaling the absolute value of the test statistic to [0,1] and then taking 1 minus the value as shown in the equation below:

$$x_{transformed} = \frac{|x| - |x_{minimum}|}{|x_{maximum}| - |x_{minimum}|} \tag{2}$$

where $x$ is the raw value before transformation. The transformation is applied on the KDE scores obtained from all methods across all datasets, thus the $x_{minimum}$ and $x_{maximum}$ are defined based on those values. The purpose of the transformation is to follow the principle of, the higher the value the better and enable easier interpretation.

To assess the validity of the KDE statistic and compare it against other measures, for example, the well-established KS test for univariate distribution, we utilised the measures implemented in countsimQC package. It includes the implementation of the following six measures: Average silhouette width, average local silhouette width, nearest neighbour (NN) rejection fraction, K-S statistics, scaled area between empirical cumulative distribution functions (eCDFs) and Runs statistics. For ease of comparing between the six measures and with the KDE test, we applied transformation rules where applicable such that the outputs from all measures are within the range of 0–1, where value closer to 1 indicates greater similarity. Similarly, the transformation is calculated from all methods across all datasets.

The measures and their transformation rules are:

1. Average silhouette width

   For each feature, the Euclidean distances to all other features were calculated. The feature was either gene or cell, depending on the data properties evaluated. A silhouette width $s(i)$ was then calculated using the following formula:

   $$s(i) = \frac{b(i) - a(i)}{\max(a(i), b(i))} \tag{3}$$

   where $b(i)$ is the mean distance between feature $i$ and all other features in the simulation data, $a(i)$ is the mean distance between feature $i$ and all other features in the original dataset.
   $s(i)$ of all features is then averaged to obtain the average silhouette width. The range of silhouette width is $[-1, 1]$. A positive value close to 1 means the data point from the simulation data is similar to the original dataset. Value close to 0 means the data point is close to the decision boundary between the original and simulated. A negative value means the data point from the original dataset is more similar to the simulation data. The same transformation as described in Eq. (2) was applied.

2. Average local silhouette width

   Similar to the average local silhouette width. The difference is that instead of calculating the distance with all the features, only the $k$ NNs were used in the calculation. Default setting of $k$ of 5 was used. The same transformation as described in Eq. (2) was applied.

3. NN rejection fraction

   First, for each feature the $k$ NNs were found using Euclidean distance. A chi-square test was then performed with the null hypothesis being the composition of $k$ NNs belonging to original and simulation data is similar to the true composition of real and simulation data. The NN rejection fraction was calculated as the fraction of features for which the test was rejected at a significance level of 5%.
   The output is the range of [0,1], where a higher value indicates greater dissimilarity between the features from real and simulation data. The value was thus transformed by taking 1 minus the value.

4. Kolmogorov-Smirnov (K-S) statistic

   The K-S measure is based on K-S statistic obtained from performing K-S test, which measures the absolute max distance between the eCDFs of simulated and real dataset. The K-S statistics is in range [0, Inf]. The K-S measure was obtained by log-transformation followed by the transformation rule defined previously.

5. Scaled area between empirical cumulative distribution functions (eCDFs) The difference between the eCDFs of the properties in simulated and real dataset. The absolute value of the difference was then scaled such that the difference between the largest and smallest value becomes 1. The area under the curve was calculated using the Trapezoidal Rule. The final value is in the range of [0,1], where a value closer to 1 indicates greater differences between the data properties distributions of the real and simulation data. The value was then reversed by taking 1 minus the value such that it follows the general pattern of higher value being better.

6. Runs statistics. The Runs statistics is the statistic from a one-sided Wald-Wolfowitz runs test. The values from the simulated and real dataset were ordered and a runs test was performed. The null hypothesis is that the sequence is a random sequence with no clear pattern of values from simulated or real dataset next to each other in position.

**Methods comparison through multi-step score aggregation**. In order to summarise the results from multiple datasets and multiple criteria, we implemented the following multi-step procedure to aggregate the KDE scores.

First, we aggregated the KDE scores within each dataset. For most methods, each cell type in a dataset containing multiple cell types was simulated and evaluated separately for the reason mentioned in the previous section. This resulted in multiple KDE scores for a single dataset, one for each cell type. To aggregate the scores into a single score for a dataset, we calculated the weighted sum by using the cell type proportion as weight, defined as the follows:

$$\sum_{i=1}^{n}(x_i * w_i) \tag{4}$$

where $n$ is the number of cell types in the simulated or original datasets, $x_i$ is the evaluation score of the $i$th cell type and $w_i$ is the cell type proportion of the $i$th cell type.

Since each method was evaluated using multiple datasets, we then summarised the performance of each method across all datasets by taking the median score. This resulted in a single score for each method on each criterion, which then enabled us to readily rank each method by comparing the score. Cases where a method was not able to produce result on particular dataset were not considered in the scoring process. The reasons for failing to simulate a data include not completing the simulation in the given time limit, error arising in the simulation methods during the simulation process, and special cases in which the simulation method is limited to an input dataset containing two or more cell types and cannot generate result on datasets containing a single cell type. The breakdown of the number of datasets successfully simulated and the number of failed cases are reported in detail in Supplementary Fig. 4.

Finally, the overall rank of each method was computed by firstly calculating its rank for each criterion and then taking the mean rank across all criteria.

**Evaluation of biological signals**. The five categories of biological signals evaluated in this study were adapted from[29] and their descriptions are detailed below.

1. DE (limma) This is the typical differentially expressed genes. Limma[30] was performed to obtain the log fold change associated with each gene. We selected genes with log2 fold change > 1.

2. DE (DEsingle) This finds the differentially expressed genes using a DE detection method DEsingle[31] that is specifically designed for scRNA-seq data.

3. DV DV stands for differentially variable genes. Bartlett's test for differential variability was performed to obtain the P-value associated with each gene.

4. DD DD stands for differentially distributed genes. K-S test was performed to obtain the P-value associated with each gene.

5. DP DP is defined as differential proportion genes. We considered genes with log2 expression greater than 1 as being expressed and otherwise as non-expressed. A chi-square test was then performed to compare the proportion of expression of each gene between two cell types.

6. BD BD is defined as bimodally distributed genes. Bimodality index defined using the below formula was calculated for each gene:

$$BI = \frac{|m_1 - m_2|}{s\sqrt{p(1-p)}} \tag{5}$$

where $m_1$ and $m_2$ are the mean expression of genes in the two cell types, respectively, $s$ is the standard deviation and $p$ is the proportion of cells in the first cell type.

For the first five categories, genes with P-value < 0.1 (Benjamini-Hochberg adjusted) were selected. This higher threshold was used instead of the typical threshold of 0.05 to result in a higher proportion of biological signals, as larger

value would enable clearer differentiation of methods' performance. For the last category, we used bimodality index[32] >0.03 as the cut-off to yield a reasonable proportion of BD genes (Supplementary Fig. 6).

To quantify the performance of each method, we used SMAPE[33]:

$$SMAPE = \frac{1}{n}\sum_{t=1}^{n}\frac{|F_t - A_t|}{\frac{A_t+F_t}{2}} \tag{6}$$

where $F_t$ is the proportion of biological signals in simulated data and $A_t$ is the proportion in the corresponding real data, $n$ is the number of data points, one from each dataset evaluated. The proportion was calculated as the number of biological signal genes divided by the total number of genes in a given dataset.

**Evaluation of scalability**. To reduce potential confounding effect, we measured scalability using the Tabula Muris dataset only. The dataset was subset to the two largest cell types and random samples of the cells without replacement were taken to generate datasets containing 50, 100, 250, 500, 750, 1000, 1250, 1500, 2500, 3000, 4000, 6000 and 8000 cells with equal proportion of the two cell types.

Running time of each method was measured using the Sys.time function built-in R and the time.time function built-in Python. Tasks that did not finish within the given time limit are considered as no result generated. To record the maximal memory for R methods we used the function Rprofmem in the built-in utils Package in R. For Python methods we used the psutil package and measured the maximal Resident Set Size. All measurements were repeated three times and the average was reported.

In the majority of methods, simulation was performed in a two-step process. In the first step, a range of properties is estimated from a given dataset. This set of properties are then used in the second step of generating the simulation data. For these methods, the time and memory usage of the two steps was recorded separately and shown in Supplementary Fig. 4. For other methods where the two processes were completed in one single function, we measured the time and memory usage of this single step and used a dashed line to indicate these methods in Supplementary Fig. 4.

In order to compare and rank the methods as shown in Fig. 2, we summed the time and memory of the methods that use two-step procedure and displayed the total time and memory usage, such that their results became comparable with methods that involve one single step. Some methods did not complete the simulation within the given time, and the time and memory usage were unable to be recorded as the result. These timed out simulations would bias the result when ranking the methods based on the total time and memory usage. To account for this case, we assigned these simulation jobs a total time usage as the time limit and a memory usage as the memory of the previous simulation task. For example, a method that failed to simulate 8000 cells within the time limit of 8 h was assigned 8 h as the total time usage, and a memory usage as the memory recorded when simulating the previous job of 6000 cells.

**Evaluation of impact of data characteristics**. We selected a subset of datasets to examine the impact of the number of cells and sequencing technologies. Briefly, each dataset was split into 50% training and 50% testing. Transformed KDE score was then calculated from the raw score obtained from all methods across the selected datasets, resulting in values ranging between 0 and 1.

*Impact of number of cells*. To assess the impact of the number of cells on the accuracy of data property estimation, we used the Tabula Muris dataset subset to the two largest cell types and sampled to create datasets of 100, 200, 500, 1000, 1500, 2000, 2500, 3000, 5000, 6000, 8000, 12,000 and 16,000 cells. Each dataset was split into 50% training and 50% testing as previously described.

Linear regression was fitted using the lm function in the built-in stats package in R for each of the 13 data properties. This resulted in a total of 13 regression models with the formula defined as:

$$y = \beta_0 + \beta_1 x_1 \tag{7}$$

The response variable $y$ was the KDE score corresponding to the data property and the exploratory variables $x_1$ was the number of cells measured in 1000.

*Impact of the sequencing protocols*. To assess the impact of the sequencing protocols while avoiding potential batch effect, we utilised two sets of datasets from the same study[20] that sequenced the same tissue type using multiple protocols. It contains human PBMC data generated using the following six protocols, 10x Genomics, CEL-seq2, Drop-seq, inDrops, Seq-Well and Smart-seq2 and mouse cortex cells using the following four protocols of sci-RNA-seq, 10x Genomics, DroNc-seq and Smart-seq2.

ANOVA was fitted using the built-in stats package in R to examine whether there was significant change in mean KDE score across the above datasets of different sequencing technologies for each simulation method. P-values were displayed on the figures.

**Reporting summary**. Further information on research design is available in the Nature Research Reporting Summary linked to this article.

## Data availability

All datasets used in this study are publicly available. Details on each dataset including accession numbers and source websites are listed in Supplementary Data 1. Curated version of the datasets is available as a Bioconductor package under the name SimBenchData (https://bioconductor.org/packages/devel/data/experiment/html/SimBenchData.html).

## Code availability

The benchmark framework is available as an R package at https://github.com/SydneyBioX/SimBench[34]. A Shiny application for interactively exploring the results is available at http://shiny.maths.usyd.edu.au/.

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

## Acknowledgements

The authors would like to thank all their colleagues, particularly at The University of Sydney, School of Mathematics and Statistics, for their intellectual engagement and constructive feedback. This study was made possible in part by the Australia National Health and Medical Research Council (NHMRC) Investigator Grant (APP1173469) to P.Y.; NHMRC CRE Grant (APP1135285) to J.Y.H.Y.; AIR@innoHK programme of the Innovation and Technology Commission of Hong Kong to P.Y. and J.Y.H.Y.; Research Training Program Tuition Fee Offset and University of Sydney Postgraduate Award Stipend Scholarship to Y.C.

## Author contributions

J.Y.H.Y. and P.Y. conceived the study. Y.C. performed the experiments and interpreted the results with input from J.Y.H.Y. and P.Y. All authors wrote, read and approved the final paper.

## Competing interests

The authors declare no competing interests.
