## [Peer Review File. · Nature Communications]

A benchmark study of simulation methods for single-cell RNA sequencing dataReviewers' Comments:

Reviewer #1:

Remarks to the Author:

I think the ms is interesting, well written, well organized and sound.

There are some methodological points that, in my opinion, need clarification.

Major issues

1. The authors divide the datasets in training and test set; use the first to learn the parameters and the second to assess the methods performance. This is fine with the exception, I think, of library size, which should be directly estimated from the test for a number of the presented methods since this is somehow a parameter related to the sequencing depth on which there is some control and the method assume this can be directly set as input. I think this is a major point because this choice can strongly affect results shown in Figure 2.

2. I think that reporting the library size, the TMM and then the effective library size as a product of the first 2 is somehow redundant. Aren't you checking twice the same properties?

3. Another important point is the ability of methods to consider different clusters of cells within the same dataset. The authors state they generate the data and estimate the parameters separately for each cluster of cell to address this issue. I think this choice helps the methods that have this limitation in particular when the expression variance and the gene/cell correlation are monitored. I think it would be fair to highlight this limitation in Figure 2.

4. In figure 3 and in the main text the authors discuss the impact of dataset characteristics on method performance. However, I would like to see (in supplementary of course) how different simulators perform on different data types. Are there simulators that perform differently on different technologies?

5. In section Methods, the authors state that for methods that requires normalized data they converted the raw counts in $\log_2 \text{CPM} + 1$. Might this be suboptimal for some methods? Do the methods suggest other strategies for normalizing input data? Can the authors report which methods requires normalized data as input?

6. Transformation (1) pg. 19 is done for sake of easier comparison among different metrics. I think the authors should specify which is the set of x on which min and max are calculated (across datasets? Across all methods?)

7. The authors also mention there are cases were some methods were not able to produce result on particular dataset (pg 23). What does this means? How many times did it happen for different methods? Can you report this aspect?

8. About the DE. Why using Limma and $\text{LFC} > 1$ in DE test? How much would different approaches affect the result?

Minor points

1. Pg.7. the authors state that , together with DE, they are assessing four other types of gene signals. I would anticipate briefly which kind of signals they are going to monitor

2. Pg 8. The authors mention that ZINB-WAVE performs poorly in terms of computational scalability. Which are the methods that perform well? Similarly, in pg.14 the authors mention the poor performers rather than the good ones.

3. Average silhouette width, local silhouette width, NN rejection fraction, KM test and eCDF are used only to assess that KDE test is a good score. Am I right? If this is the case I would report these alternative assessment scores in supplementary

Reviewer #2:

Remarks to the Author:

This is a very timely and well performed study. I have no substantial comments. I would only have liked to see a bit more information about the R package. From the paper it is not clear what users can do with the R package, apart from it containing all datasets used in the paper.

Could you also say a bit more about how to go from here? I mean, if in the future new simulation methods will be developed, can these methods be easily added to your comparative study? Is this what the R package can be used for?

Reviewer #3:

Remarks to the Author:

Specific comments:

A number of methods have been developed specifically for simulating single-cell RNA-seq (scRNA-seq) data: Splatter, SymSym, SPSimSeq, cscGAN, SERGIO, PROSSTT (although perhaps not as relevant here as PROSSTT simulates datasets for differentiation processes). The authors evaluate most of these methods here (SERGIO and PROSSTT are not evaluated - the authors should at least comment on why these are omitted).

In addition to these methods, a number of other simulation approaches are evaluated, even though they were not designed specifically to simulate scRNA-seq data. The approach referred to as powsimR (since it came from the powsimR paper) is a good example. PowsimR was proposed as a tool to assess power and sample size requirements for differential expression (DE) analysis in single cell and bulk RNA-seq experiments. The authors of the powsimR paper note that the negative binomial model provides an adequate fit for 54% of the genes from non-UMI-methods and 39% of the genes for UMI-methods. The powsimR tool uses the NB model as a default, but allows a user to simulate data from a zero inflated NB model. The authors are focused on the particular question of estimating power and sample size, not on providing users with simulated data to use in general evaluations. As such, this approach should not be included in this paper. Another example is scDD. In that work, the authors propose scDD as an approach for identifying genes that are differentially distributed (DD) across two conditions. They focused on fluidigm data, and filtered out genes having 90% or more zeros. Their approach is not appropriate for more recent scRNA-seq technologies given the high sparsity, and as with powsimR, their simulation was designed to address very specific questions associated with their approach, not as a general purpose tool. SparseDC is another example. SparseDC identifies cell types, traces their changes across conditions and identifies associated marker genes. As with scDD, the simulation there was designed to evaluate the operating characteristics of SparseDC, not as a general purpose simulation tool. These approaches (powsimR, scDD and SparseDC - and perhaps others - should be removed).

The authors conclude that it's unclear which model affects performance because the methods they identify at the top are all using different models, and that some methods with similar models performed quite differently. This seems to be an extremely important question that should be well understood by the authors and clearly addressed in the paper.

Similarly, the authors note that cscGAN " only had moderate performance...We speculate that this could be due to the sample size required to train a deep learning model in general." This speculation should be addressed and clarified.

The authors of cscGAN show that their approach outperforms Splatter on many metrics, which seems inconsistent with the results reported here. Some discussion is warranted.

How different are the ranks that order methods? Specifically, are two methods ranked 1 and 2 on some metric significantly different in some way? Overall?

By aggregating scores over all datasets, it is difficult to identify which methods perform well in certain cases (e.g. datasets with lots of cell types).

Minor comment:

The authors note that "Limma was performed to obtain the log fold change associated with each gene. We selected genes with log fold change > 1." Do you mean $\log_2 > 1$?

We thank the positive review from all reviewers. We have addressed all comments raised by each of the reviewers point by point below.

Reviewer #1 (Expertise: bioinformatics, scRNASeq data simulation):

I think the ms is interesting, well written, well organized and sound. There are some methodological points that, in my opinion, need clarification.

Major issues

1. The authors divide the datasets in training and test set; use the first to learn the parameters and the second to assess the methods performance. This is fine with the exception, I think, of library size, which should be directly estimated from the test for a number of the presented methods since this is somehow a parameter related to the sequencing depth on which there is some control and the method assume this can be directly set as input. I think this is a major point because this choice can strongly affect results shown in Figure 2.

Response: We noted that there are only three methods (Zinger, scDesign and SPsimSeq) that have library size as an input parameter in the simulation function. As suggested, we have now re-run Zinger with the library size sampled from the test data and re-run scDesign with the sequencing depth set to that of the test data instead of using the default number. We note that, for SPsimSeq, the default behaviour of the method is to estimate the library size parameter by fitting a log-normal distribution for the input data. We kept the simulation results from using the default behaviour of the package.

We observed that the ranking of zinger remained the same, while the ranking for scDesign slightly improved (now better than zinger and POWSC). The overall tier ranking of all 12 simulation methods were not impacted (ie, tier 1, 2 and 3 contains the same methods as before). We have revised Figure 2, Figure 3 and Figure 4 to reflect these changes.

2. I think that reporting the library size, the TMM and then the effective library size as a product of the first 2 is somehow redundant. Aren't you checking twice the same properties?

Response: These metrics were used in the countsimQC package (Ref. 1) and adopted in this study, as described in the Methods section of the manuscript. In particular, library size was used to examine the distribution of the read counts, whereas TMM (Ref. 2), the normalisation factor of library size, is calculated from the library size after outlier removal. Therefore these metrics do capture different data properties. The following two examples compare the distribution of the real data and the simulated data generated by SymSim. The first example shows that SymSim simulated data has a relatively similar distribution to the read data in terms of TMM but has quite a different distribution in terms of library size and effective library size. In comparison, the second example shows that SymSim simulated data has relatively similar distributions in library size and TMM compared to read data, but the discrepancy in distribution is apparent in terms of effective library size. As such, we have kept all metrics as they offer different perspectives.

Ref 1. Sonesson, C., & Robinson, M. D. (2018). Towards unified quality verification of synthetic count data with countsimQC. *Bioinformatics*, 34(4), 691-692.

Ref 2. Robinson, M. D., & Oshlack, A. (2010). A scaling normalization method for differential expression analysis of RNA-seq data. *Genome biology*, 11(3), 1-9.

3. Another important point is the ability of methods to consider different clusters of cells within the same dataset. The authors state they generate the data and estimate the parameters separately for each cluster of cell to address this issue. I think this choice helps the methods that have this limitation in particular when the expression variance and the gene/cell correlation are monitored. I think it would be fair to highlight this limitation in Figure 2.

Response: We thank the reviewer for raising this aspect and have now expanded our evaluation framework from three aspects of criteria to four by adding an aspect of criteria termed “applicability” that specifically addresses this point (see revised Figure 2). This new category of criteria focusing on “applicability” examines three criteria, (i) the ability of methods to estimate parameters from multiple cell types, (ii) to simulate data with multiple number of cell types and (iii) to control the amount of differential expression.

Accordingly, we have updated the result section of the manuscript and the following paragraph has been added on Page 9 describing the results from the “applicability” criteria.

“Lastly, we found that different simulation methods satisfy different numbers of the applicability criteria (Fig. 2d). This is due, in part, to the fact that not all simulation methods are designed as general purpose simulation tools. For example, powsimR was originally designed as a power analysis tool for differential expression analysis but was included as a simulation tool by a number of simulation studies (Li and Li 2019) (Zhang et al. 2019) in their performance comparison with other simulation methods. Being a power analysis tool, its primary usage is to simulate two cell groups from a homogenous cell population with a user-defined amount of differential expression. In contrast, a number of other methods such as SPARSim, SymSim and Splat that are originally intended as general purpose simulation tools are able to simulate multiple cell groups with user-defined differential expression patterns. We have outlined the primary purpose and the limitations of each method on this front in more detail in Table 1 to guide users in making informed decisions on methods that best suited to their needs.”

We believe this additional aspect of criteria and the associated discussion will now better inform readers about the designed capabilities and limitations of each method.

4. In figure 3 and in the main text the authors discuss the impact of dataset characteristics on method performance. However, I would like to see (in supplementary of course) how different

simulators perform on different data types. Are there simulators that perform differently on different technologies?

Response: We have added a new figure, Supplementary Figure 7, to show the breakdown of each method on each dataset that was used to assess the impact of sequencing technologies (six protocols on human PBMC and four protocols on mouse Cortex). We assessed whether there is a technology (dataset) effect on each method (based on ANOVA) and reported the p-value on the supplementary figure. Based on the ANOVA results, we found all methods perform similarly across all technologies, consistent with our original results.

5. In section Methods, the authors state that for methods that requires normalized data they converted the raw counts in $\log_2 \text{CPM} + 1$. Might this be suboptimal for some methods? Do the methods suggest other strategies for normalizing input data? Can the authors report which methods requires normalized data as input?

Response: Among all simulation methods, Sparsim is the only one that requires normalised data in addition to raw data. Sparsim recommends using Scran normalisation while noting other normalisation procedures could also be used as well. We have now re-ran Sparsim using Scran normalisation and observed it did not affect the rank of Sparsim (see revised Figure 2). Other relevant figures (Figure 3 and Figure 4) have also been updated to reflect this change.

6. Transformation (1) pg. 19 is done for sake of easier comparison among different metrics. I think the authors should specify which is the set of x on which min and max are calculated (across datasets? Across all methods?)

Response: We have now specified this in the Methods section on Page 21 as shown below.

“The transformation is applied on the KDE scores obtained from all methods across all datasets, thus the x_{minimum} and x_{maximum} are defined based on those values.”

7. The authors also mention there are cases were some methods were not able to produce result on particular dataset (pg 23). What does this means? How many times did it happen for different methods? Can you report this aspect?

Response: We have now added a figure showing the percentage of times that a method failed to produce simulation results (Supplementary Figure 4). We have also clarified this in the relevant Methods section on Page 25 as shown below.

“The reasons for failing to simulate a data include not completing the simulation in the given time limit, error arising in the simulation methods during the simulation process, and special cases in which the simulation method is limited to an input dataset containing two or more cell types and cannot generate result on datasets containing a single cell type. The breakdown of the number of datasets successfully simulated and the number of failed cases are reported in detail in Supplementary Fig. 4.”

8. About the DE. Why using Limma and $\text{LFC} > 1$ in DE test? How much would different approaches affect the result?

Response: We acknowledge that the choice of DE test methods would give different ranking of DE genes and may lead to different evaluation results of the simulation methods. As suggested, we have now also included DEsingle, a method designed specifically for differential expression analysis of scRNA-seq data (Ref. 3). We found that the ranking of the simulation methods based

on DEsingle correlates closely with those from limma (see figure below). We have now revised the manuscript including these new results (see revised Figure 2b and revised Figure 3b) and added the following text on Page 16 in the Discussion section as shown below.

“Although the choice of DE detection methods could affect the evaluation of the simulation methods, our evaluation using both limma, a DE method originally designed for bulk RNA-seq data, and DEsingle, a DE method specifically designed for scRNA-seq data demonstrate a high agreement of the rankings of simulation methods based on the two DE methods (Fig 2b).”

Ref. 3: Miao, Z., Deng, K., Wang, X., & Zhang, X. (2018). DEsingle for detecting three types of differential expression in single-cell RNA-seq data. *Bioinformatics*, 34(18), 3223-3224.

Minor points

1. Pg.7. the authors state that , together with DE, they are assessing four other types of gene signals. I would anticipate briefly which kind of signals they are going to monitor

Response: We have edited the relevant section and listed the types of gene signals and added the following on Page 7 in the Results section.

“For biological signals, we measured the proportion of differentially expressed (DE) genes obtained in the simulated data using DE detection methods designed for bulk and single-cell RNA-seq data, as well as four other types of gene signals of differentially variable (DV), differentially distributed (DD), differential proportion (DP) and bimodally distributed (DP) genes (see Methods).”

2. Pg 8. The authors mention that ZINB-WAVE performs poorly in terms of computational scalability. Which are the methods that perform well? Similarly, in pg.14 the authors mention the poor performers rather than the good ones.

Response: The original discussion in Page 14 focused on aspects of future development. ZINB-WaVE is the top performer in parameter estimation but a poor performer in scalability, and we mentioned this method to highlight the challenge and the needs of constructing an accurate model whilst maintaining scalability. We agree that it is important to identify the good performers and have now revised the manuscript and mentioned methods that are top performers in both aspects of criteria. Specifically, we added the following on Page 9 in the Results section.

“In contrast, methods such as SPARSim, which was ranked second in parameter estimation as well as being one of the top tier methods in scalability, may better suit needs if a large-scale simulation dataset is required by users.”

We also added the following discussion on Page 15 in the Discussion section.

“For example, as single-cell field advances and datasets with hundreds of thousands of cells become increasingly common, users may be interested in simulating large-scale datasets to test the scalability of their methods. As a result, methods that rank highly on scalability while also performing well on other aspects (e.g., SPARSim, scDesign and Splat) may be more favourable than other methods under these scenarios.”

3. Average silhouette width, local silhouette width, NN rejection fraction, KM test and eCDF are used only to assess that KDE test is a good score. Am I right? If this is the case I would report these alternative assessment scores in supplementary.

Response: Given the scores are measured across each parameter, each dataset and each simulation method, resulting in thousands of measurements for each metric, we have resorted to reporting pairwise correlation of the seven assessment metrics and updated them in the revised Supplementary Figure 3. These results demonstrate that the seven metrics are highly correlated.

Reviewer #2 (Expertise: bioinformatics, biostatistics, scRNAseq data analysis):

This is a very timely and well performed study. I have no substantial comments. I would only have liked to see a bit more information about the R package. From the paper it is not clear what users can do with the R package, apart from it containing all datasets used in the paper. Could you also say a bit more about how to go from here? I mean, if in the future new simulation methods will be developed, can these methods be easily added to your comparative study? Is this what the R package can be used for?

Response: We thank the reviewer for the positive comments and suggestions. We have developed two packages (1) SimBench, containing the evaluation framework, and (2) SimBenchData, containing the datasets used in the paper. These two packages work together to enable assessment of future methods within the proposed framework. As part of our strong belief in open source research, these packages have been submitted to the Github and Bioconductor repository. We have now clarified this on Page 16 in the Discussion section of the revised manuscript as shown below.

“The evaluation framework has been made publicly available as the R package SimBench (<https://github.com/SydneyBioX/SimBench>). SimBench allows any new simulation methods to be readily assessed under our framework. It requires two inputs including the simulated data generated by any simulation method and the real data that was used as the reference input to generate the simulated data. SimBench then runs the evaluation procedure as performed in this study. We also provide all datasets used in this study as a bioconductor data package SimBenchData (<https://bioconductor.org/packages/devel/data/experiment/html/SimBenchData.html>). Together these two packages enable future simulation methods to be assessed and compared with the methods benchmarked in this study.”

In addition, we have developed a Shiny application hosted on the website (<http://shiny.maths.usyd.edu.au/>) that allows users to explore the benchmark results in an interactive

manner. The website will be continuously maintained and updated to include benchmark results from new simulation methods when they become available or when we are informed by methods developers. As such, our comparative study fosters future development.

Reviewer #3 (Expertise: Statistics, Genomics):

Specific comments:

A number of methods have been developed specifically for simulating single-cell RNA-seq (scRNA-seq) data: Splatter, SymSym, SPSimSeq, cscGAN, SERGIO, PROSSTT (although perhaps not as relevant here as PROSSTT simulates datasets for differentiation processes). The authors evaluate most of these methods here (SERGIO and PROSSTT are not evaluated - the authors should at least comment on why these are omitted).

Response:

As pointed out by the reviewer, methods such as PROSSTT and dyngen are designed for simulating differentiation processes and thus require lineage tree in its input, whereas SERGIO simulates regulation of genes by transcriptional factors and requires user defined gene regulatory network. We agree with the reviewer that it is worthwhile to mention these methods so that readers are aware there are simulation methods designed for very specific purposes. We have now added a section in the discussion outlining these methods and further clarify our inclusion and exclusion criteria behind this benchmarking study. We have added the following text on Page 16 in the Discussion section.

“While we aim to provide a comprehensive assessment of available simulation methods, our study is not without limitations. For example, a few methods were excluded in this study due to their unique properties. SERGIO²³ is able to simulate regulation of genes by transcriptional factors, and therefore requires gene regulatory networks as one of the inputs. Both PROSSTT²⁴ (Papadopoulos et al., 2019) and dyngen²⁵ (Cannoodt et al., 2021) are designed to simulate scRNA-seq data with trajectory information and require user-defined lineage trees. Lun²⁶ (Lun and Marioni 2017) was originally designed to tackle confounding plate effects in DE analysis and it requires plate information to be specified in the input. These simulation methods may need special consideration and evaluation criteria that could not be captured by the general framework in this study.”

In addition to these methods, a number of other simulation approaches are evaluated, even though they were not designed specifically to simulate scRNA-seq data. The approach referred to as powsimR (since it came from the powsimR paper) is a good example. PowsimR was proposed as a tool to assess power and sample size requirements for differential expression (DE) analysis in single cell and bulk RNA-seq experiments. The authors of the powsimR paper note that the negative binomial model provides an adequate fit for 54% of the genes from non-UMI-methods and 39% of the genes for UMI-methods. The powsimR tool uses the NB model as a default, but allows a user to simulate data from a zero inflated NB model. The authors are focused on the particular question of estimating power and sample size, not on providing users with simulated data to use in general evaluations. As such, this approach should not be included in this paper. Another example is scDD. In that work, the authors propose scDD as an approach for identifying genes that are differentially distributed (DD) across two conditions. They focused on fluidigm data, and filtered out genes having 90% or more zeros. Their approach is not appropriate for more recent scRNA-seq technologies given the high sparsity, and as with powsimR, their simulation was designed to address very specific questions associated with their approach, not as a general purpose tool. SparseDC is another example. SparseDC identifies cell types, traces their changes across conditions and identifies associated marker genes. As with scDD, the simulation there was

designed to evaluate the operating characteristics of SparseDC, not as a general purpose simulation tool. These approaches (powsimR, scDD and SparseDC - and perhaps others - should be removed).

Response: We agree with the reviewer that while all methods provided in our benchmark study enable modeling of scRNA-seq data, some of them are not intended to be general-purpose simulation tools. Nevertheless, in practice all these methods were commonly applied for data simulation. For example, both scDesign (Ref. 4) and SymSim (Ref. 5) study included powsimR in their comparison study; Splatter (Ref. 6) and scDesign (Ref. 4) both includes scDD in their comparison study; and SparseDC was included in the Splatter package as one of the simulation method choices.

To clarify this issue, we have now added an additional aspect of evaluation criteria termed “applicability” and updated Figure 2 and Table 2 in the revised study to better specify the design purposes and applicability of the included methods. Given that all included methods are frequently used by the single-cell community for data simulation, we believe it is still worthwhile to keep these methods in our study given how they are used in the community and also for raising attention/awareness on their designing purposes and applicability when they are being applied for data simulation purposes.

Ref. 4 Li, W. V., & Li, J. J. (2019). A statistical simulator scDesign for rational scRNA-seq experimental design. *Bioinformatics*, 35(14), i41-i50.

Ref. 5 Zhang, X., Xu, C., & Yosef, N. (2019). Simulating multiple faceted variability in single cell RNA sequencing. *Nature communications*, 10(1), 1-16.

Ref. 6 Zappia, L., Phipson, B., & Oshlack, A. (2017). Splatter: simulation of single-cell RNA sequencing data. *Genome biology*, 18(1), 1-15.

The authors conclude that it's unclear which model affects performance because the methods they identify at the top are all using different models, and that some methods with similar models performed quite differently. This seems to be an extremely important question that should be well understood by the authors and clearly addressed in the paper.

Response: We thank the reviewer for raising this point and have further clarified this part in our writing. Specifically, ZINB-WaVE, zingeR and powsimR all use models from the negative binomial (NB) family, but were found to be tier 1, tier 2 and tier 3 methods respectively in terms of parameter estimation. We have now clarified this point in the discussion section that the actual implementation and approaches in all three methods differ substantially in detail despite the fact that they are all inspired by representing the observed counts using the NB family. Specifically, we have added the following text on Page 13 in the Discussion section as shown below.

“Specifically, zingeR uses NB distribution to fit the mean and dispersion of the count data and model the excess zero using the interaction between gene expression and sequencing depth using additive logistic regression model. powsimR uses the standard zero inflated NB (ZINB) distribution to fit the mean and dispersion of the count data, with the zero inflation modelled using binomial sampling. In ZINB-WaVE, the ZINB distribution is used to fit the mean and dispersion of the count data, as well as the probability that a zero is observed. Additionally, the estimation of mean and zero probability incorporates an additional parameter adapted from the RUV framework²¹ to capture unknown cell-level covariates. Therefore, while both powsimR and ZINB-WaVE use ZINB distribution to fit the count data, the actual model differs.”

Similarly, the authors note that cscGAN "only had moderate performance...We speculate that this could be due to the sample size required to train a deep learning model in general." This speculation should be addressed and clarified.

Response: We commented on this point as we observed that the performance of cscGAN is sensitive to the number of cells used for training the model (Ref. 7). In particular, as shown in their downsampling experiment, a large number of cells were required to train a useful cscGAN model and the quality of the simulated data, in terms of how well they represent the real data, improves when increasingly more cells were used for training the cscGAN model (Figure 2g and Supplementary Figure 18a in Ref. 7). We have now reworded this sentence on Page 13 in the Discussion section to clarify this point as shown below.

"Interestingly, while deep learning methods have dominated various fields and applications, cscGAN, a deep learning based model, for scRNA-seq data simulation only had moderate performance compared to the other models. This may be due to the large number of cells required for training the deep neural network in cscGAN as was demonstrated in their original study (Marouf et al)."

Ref. 7: Marouf, M., Machart, P., Bansal, V., Kilian, C., Magruder, D. S., Krebs, C. F., & Bonn, S. (2020). Realistic in silico generation and augmentation of single-cell RNA-seq data using generative adversarial networks. *Nature communications*, 11(1), 1-12.

The authors of cscGAN show that their approach outperforms Splatter on many metrics, which seems inconsistent with the results reported here. Some discussion is warranted.

Response: We acknowledge that cscGAN outperformed Splatter on t-SNE visualization (Supplementary Figure 4 D-F in Ref. 7), marker gene correlation (Supplementary Figure 4 A-C in Ref. 7) and MMD (Supplementary Table 2 in Ref. 7). However, cscGAN did not outperform Splatter on five out of the six data properties examined and reported in their original study (Supplementary Figure 5 A-F in Ref. 7), including mean expression, mean variance, zeros per gene, zeros per cell, and mean count versus percentage zeros. In our study, the simulation methods were evaluated based on data properties since these criteria are interpretable, can be quantitatively evaluated and summarised across multiple datasets and applicable to all simulation methods. Therefore, we believe our benchmark result is consistent with the results reported in the cscGAN study.

How different are the ranks that order methods? Specifically, are two methods ranked 1 and 2 on some metric significantly different in some way? Overall?

Response: In Fig.4, we present the KDE scores (scaled for readability) of each method on each evaluation criterion, which are the values used to calculate the method ranks in Fig.2. To make it simpler to explore these KDE scores, we have now created a Shiny app hosted at <http://shiny.maths.usyd.edu.au/> with an interactive table that displays the values for all criteria. The app allows users to sort the KDE value by any criteria of interest, making it easy to compare how similar any two methods are on any criterion.

By aggregating scores over all datasets, it is difficult to identify which methods perform well in certain cases (e.g. datasets with lots of cell types).

Response: We agree with the reviewer and believe future developers will benefit from looking into the method performance in specific cases and criteria. The Shiny app

(<http://shiny.maths.usyd.edu.au/>) mentioned above provides the benchmark results of each of all methods on all datasets and can accommodate this need. It has the advantage of interactivity and flexibility, where users can rank the methods based on various data characteristics. Specifically, it allows selection of the datasets by number of cells, by number of cell types, or through manual selection. As the website can be updated (unlike static figures), we will update it to include benchmark results from new simulation methods when they become available so that our comparative study will stay up-to-date and will support future method development.

Minor comment:

The authors note that “Limma was performed to obtain the log fold change associated with each gene. We selected genes with log fold change > 1.” Do you mean $\log_2 > 1$?

Response: We thank the reviewer for pointing this out. Yes, we meant $\log_2 > 1$. This has now been corrected in the revised manuscript.

Reviewers' Comments:

Reviewer #1:

None

Reviewer #2:

Remarks to the Author:

Thank you very much for the answers to my questions.

Although the original submission was already of very good quality, this revision even further improved the manuscript. It is a timely and well written paper! I also like the R package and the open research approach.

Reviewer #3:

Remarks to the Author:

The reviewers have addressed all of my comments.

We thank the positive review from the reviewers and for supporting the publication of our study. There are no further comments to address as shown below.

Reviewer #2 (Remarks to the Author):

Thank you very much for the answers to my questions. Although the original submission was already of very good quality, this revision even further improved the manuscript. It is a timely and well written paper! I also like the R package and the open research approach.

Reviewer #3 (Remarks to the Author):

The reviewers have addressed all of my comments.